# The PI3K/Akt Pathway: Emerging Roles in Skin Homeostasis and a Group of Non-Malignant Skin Disorders

**DOI:** 10.3390/cells10051219

**Published:** 2021-05-17

**Authors:** Yan Teng, Yibin Fan, Jingwen Ma, Wei Lu, Na Liu, Yingfang Chen, Weili Pan, Xiaohua Tao

**Affiliations:** 1Department of Dermatology, Zhejiang Provincial People’s Hospital, People’s Hospital of Hangzhou Medical College, Hangzhou 310014, China; tengyan@hmc.edu.cn (Y.T.); fanyibin@hmc.edu.cn (Y.F.); majingwen@hmc.edu.cn (J.M.); luwei@hmc.edu.cn (W.L.); 2Graduate School of Bengbu Medical College, Bengbu 233000, China; 20186631224@stu.bbmc.edu.cn (N.L.); 20196631254@stu.bbmc.edu.cn (Y.C.)

**Keywords:** PI3K/Akt signaling pathway, skin, homeostasis, non-malignant disorders, targeted therapies

## Abstract

The phosphatidylinositol 3-kinase (PI3K)/protein kinase B (Akt) signaling pathway regulates cell proliferation, differentiation, and migration, along with angiogenesis and metabolism. Additionally, it could mediate skin development and homeostasis. There is much evidence to suggest that dysregulation of PI3K/Akt pathway is frequently associated with several human cutaneous malignancies like malignant melanoma (MM), basal cell carcinoma (BCC), and cutaneous squamous cell carcinoma (SCC), as well as their poor outcomes. Nevertheless, emerging roles of PI3K/Akt pathway cascade in a group of common non-malignant skin disorders including acne and psoriasis, among others, have been recognized. The enhanced understanding of dysfunction of PI3K/Akt pathway in patients with these non-malignant disorders has offered a solid foundation for the progress of updated therapeutic targets. This article reviews the latest advances in the roles of PI3K/Akt pathway and their targets in the skin homeostasis and progression of a wide range of non-malignant skin disorders and describes the current progress in preclinical and clinical researches on the involvement of PI3K/Akt pathway targeted therapies.

## 1. Introduction

The skin is the largest sensory organ affected by major environmental factors such as ultraviolet (UV) exposure, wounds, oxidative stress, and microbial infection, covering the surface of the body [1,2]. As to restore damaged tissues or cells and replace aging cells, various stem cell pools residing in the skin facilitate the maintenance and repair the different part of skin, including epidermis, dermis, hair follicles, etc., [3,4], which are maintained by a variety of signaling pathways like Wnt [5], transforming growth factor β (TGF-β) [6], Notch and Hedgehog (HH), PI3K/Akt pathway, etc., [7,8]. The classical PI3K/Akt pathway is related to the regulation of a variety of physiological activities, including cell proliferation, differentiation, apoptosis, angiogenesis, metabolism, and protein synthesis [9,10,11,12]. In skin, the activation of the PI3K/Akt pathway is responsible for maintaining the skin homeostasis. In addition to the skin tumors like melanoma, BCC, and SCC, dysregulation of the PI3K/Akt pathway cascade is also reported to be involved in a group of non-malignant skin disorders including acne, psoriasis, vitiligo, scleroderma, et al. [13]. In this article, we present a comprehensive and novel understanding of the emerging roles and therapeutic targets of the PI3K/Akt pathway in skin homeostasis and a group of non-malignant skin disorders.

## 2. The Skin Structure and Function

The skin is a relatively complex organ structure outside the body as shown in Figure 1, which composes of the epidermis, dermis, subcutaneous layer, as well as blood vessels and nerves. These components are combined to protect the body from trauma, environmental stimulus, and microbial infection. The skin is also viewed as the neuroendocrine organ that can generate the signals to produce rapid (neural) or slow (humoral or immune) responses at the local and systemic level exposed to hostile environments. The sensory and regulatory function of the skin are integrated into the skin immune, pigmentary, epidermal, and adnexal system, and their connection with the neuroendocrine system to maintain local and systemic homeostasis [14,15,16,17]. Ultraviolet (UV) exposure, as a key determinant factor in life, makes a great impact on the skin biology and pathology, as well as the whole organism, which could not be separated with the skin neuroendocrine capabilities. UV radiation (UVR), mainly UVB, can upregulate local neuroendocrine axes that comprise of cytokines, corticotropin-releasing hormone, urocortins, proopiomelanocortin-peptides, enkephalins, or others. They can also exert systemic effects through the circulative release, including activation of the central hypothalamic-pituitary-adrenal axis, immunosuppression, and independent of vitamin D synthesis [18]. The skin barrier generally refers to the epidermis barrier. It mainly comprises of keratinocytes with different differentiation degrees from the basal cell layer to the stratum corneum. Keratinocytes are generated and renewed by the stem cells residing in the basal layer as a result of epidermis replacement every 28 days. Keratinocytes produce keratins, structural proteins that form the epidermis cytoskeleton. Filaggrin, transformed from profilaggrin, aggregates keratin filaments into tightly compressed parallel bundles that form the matrix in the stratum corneum. Thereby, any structural damage of epidermis probably leads to skin barrier damage, which induces a various of skin disorders [19,20]. Melanocytes compose approximately 10% of cells in the basal cell layer. The general population almost have the same number of melanocytes. The variations in melanin are responsible for the different degree of skin color [21,22]. The cutaneous melanin pigment is known to protect the skin from harmful influences of solar exposure. Melanin is synthesized from tyrosine among several steps that require the enzyme tyrosinase. The mechanisms of melanogenesis regulation is relatively complex, involving the transcriptional regulation, intracellular signal transduction pathways and dual function of L-tyrosine and L-DOPA. The L-tyrosine and L-DOPA has been substantiated to serve as substrates and intermediates of melanogenesis, as well as acting as positive regulators of melanogenesis and other cellular functions. The hormonal and nutritional regulation functions in the melanogenesis consistent with the old theory that receptors or amino acid-derived hormones arose from the receptors or those amino acids, and that nuclear receptors evolved from primitive intracellular receptors binding nutritional factors or metabolic intermediates [23,24]. Recent studies concerning the role of melanogenesis in regulation of melanoma behavior had demonstrated dramatic changes in the cells metabolism both on biochemical and on molecular levels, which were accompanied with dramatic stimulation of HIF-1a and HIF-independent attendant pathways [25,26]. Vitiligo is associated with melanocyte deficiency. Multiple theories have been proposed to explain melanocyte destruction, including genetics basis, autoimmunity, melanocyte self-destruction hypothesis, oxidative stress hypothesis, neural hypothesis, and melanocytorrhagy hypothesis. Among these, autoimmunity and oxidative stress hypothesis are best supported by existing research [27,28].

The dermis varies in thickness between 1 and 4 mm, which comprises of abundant cells (fibroblasts, histiocytes, etc.), collagen fibers, reticular fibers, elastic fibers, blood vessels, hair follicles, sweat, and sebaceous glands that nourish and support the epidermis and subcutaneous layer. Fibroblasts is responsible for collagen synthesis. The quantity and functional changes of fibroblasts and collagen might result in a group of skin disorders like keloid and scleroderma [29].

The subcutaneous layer located underneath the dermis is composed of adipose tissue, blood vessels, nerves, and cutaneous appendages like sweat glands, sebaceous glands, and hair follicles, which helps the body regulate temperature. The sebaceous glands that mainly comprise of sebocytes secret sebum, which moisturize the skin’s surface and inhibit microbial reproduction. Normal secretion of sebocytes is necessary to maintain skin homeostasis, mainly regulated by the level of androgen. Excessive sebum secretion is associated with the pathogenesis of acne or seborrheic dermatitis. Epithelial–mesenchymal interactions are crucial for hair follicle development and growth, which could simply be divided in into three phages: initiation, organogenesis, and cytodifferentiation. Various types cells and tissues are involved in the three stages. Once the homeostasis of hair follicles is broken, it might result in a series of hair disorders.

## 3. The PI3K/Akt Signaling Pathway

According to its function and structure, PI3K could be divided into three types, of which the most widely investigated is the type I PI3K. It is a heterodimer composed of a catalytic subunit and a regulatory subunit. The regulatory subunits contain SH2 and SH3 domains that bind with target proteins with appropriate sites. The subunit mentioned above is usually called p85, referring to the first isotype discovered, and there currently exists six known regulatory subunits, ranging in size from 50 to 110 kDa. Additionally, there exists four categories of catalytic subunits, namely p110α, β, δ, and γ, respectively. The δ is confined to leukocytes, and others are widely distributed in different cells [30].

As is shown in Figure 2, the serine/threonine kinase Akt is a proto-oncogene with functions that regulate different cell activities, including proliferation, growth, survival, apoptosis, metabolism, transcription, and protein synthesis. The components that can activate the Akt signal cascade, including receptor tyrosine kinases, integrins, B cell and T cell receptors, cytokine receptors, G protein-coupled receptors, and other phosphatidylinositol three Kinase (PI3K), elicit a stimulus to produce (3,4,5) phosphatidylinositol triphosphate (PIP3). The PIP3, transformed from PIP2 by the stimulus of PI3K, could activate the Akt signal cascade. PI3K-related kinase (PIKK) family members like DNA-PK can also phosphorylate Akt at the site of serine 473. The Akt could also be dephosphorylated by the protein phosphatase 2A (PP2A) and PH-domain rich leucine-repeat-containing protein phosphatase (PHLPP1/2) [31,32].The PTEN is a crucial upstream molecule of the PI3K/Akt pathway, which could inhibit cell proliferation and enhance cell sensitivity to apoptosis. The major substrate of PTEN is PIP3, which can dephosphorylate PIP3 at site D3 to generate PIP2, and negatively regulate the PI3K/Akt pathway. The inactivation of PTEN elicits the continuous activation of PI3K/Akt.

Additionally, PI3K/Akt binds the various downstream molecules to function in cell motility. The Bcl-2 family members like Bcl-2-associated death promoter (BAD) are responsible for the regulation of cell apoptosis. Akt down-regulates the BAD to inhibit cell apoptosis and promote cell survival [33]. Akt can also directly phosphorylate and inactivate Caspase-9 at the site S196 and inhibit Caspase-9-mediated cell apoptosis. The mTORC1 regulates the translation initiation and ribosome synthesis, then promotes cell growth and proliferation [34,35]. The Akt could directly phosphorylate the mTOR2448 to then activate mTORC1. It can also directly phosphorylate TSC2 S939 and T1462 to inhibit TSC2 function, and then indirectly activate mTORC1 through Rheb-GTP. The Akt activates endothelial cell growth factor and phosphorylates endothelial NO synthase S1177 to increase the production of NO of endothelial cells and then stimulate the growth and proliferation of endothelial cells, increase vascular permeability, and promote angiogenesis. The downstream protein p70S6K of mTOR can promote cell movement after activation, The PI3K/Akt can also up-regulate the mRNA expression of matrix metalloproteinase-2(MMP-2), which can degrade the extracellular matrix and promote cell invasion and metastasis.

The forkhead box O (FOXO) belongs to the transcription factor family, which is an essential downstream factor of the PI3K/Akt pathway [36]. The members of the mammalian FOXO family include FOXO1, 3, 4, and 6, which have highly similar structure, function, and regulation. They mainly differ in tissue expression. In the nucleus, FOXOs mediate a wide range of transcription of target genes associated with cellular physiological events, including apoptosis, cell-cycle control, glucoses metabolism, oxidative stress resistence, wound healing, and longevity [37]. Thereby, the FOXOs proteins are also involved in the pathogenesis of several skin disorders like acne, psoriasis, etc.

## 4. Emerging Roles in Skin Homeostasis

### 4.1. The PI3K/AKT Pathway Is Necessary to Maintain the Epidermal Barrier Function

The epidermal tissue is the primary environmental barrier that protects against damage from pathogens, allergens, and ultraviolet exposure. The terminally differentiated, anuclear keratinocytes act as the major factor constituting the epidermal barrier [38]. Mice without the Akt1 and Ak2 isoforms have no stratum corneum and die neonatally, possibly due to the defects of the barrier, and Akt has been identified to have a function in the differentiation and survival of keratinocytes [39]. Heat shock proteins B1(HspB1), also known as Hsp27, is a well-established Akt substrate [40,41]. It was found that Akt-mediated phosphorylation of HspB1 elicits a transient interaction with filaggrin and intracellular redistribution. The filaggrin acts as a key protein in the formation of stratum corneum and is essential for maintaining the function of the epidermal barrier. Additionally, Akt signaling increased as the barrier wave crossed the epidermis and Jun was then transiently dephosphorylated several days before birth [42]. The acquisition of a developmental barrier was regulated by Pp2a regulation of Jun dephosphorylation and downstream of Akt signaling [43]. Manar et al. [44] discovered that temporal deficiency of pelota protein contributes to neonatal lethality before the acquisition of an epidermal barrier as the result of perturbations in permeability barrier formation. It is a corresponding outcome of failure of processing profilaggrin into filaggrin monomers, which can promote the constitution of a protective epidermal layer outside the body. They also found that pelota protein functions as a negative component to regulate the activities of the PI3K/Akt pathway in the epidermis. Furthermore, increased activity of the PI3K/Akt signaling pathway in skin deficient in pelota might impact the dephosphorylation of profilaggrin, which in turn influences its correct evolvement into filaggrin monomers and eventually epidermal differentiation.

### 4.2. Activation of the PI3K/Akt Pathway Could Induce Hair Follicle Regeneration by Promoting the Differentiation and Proliferation of HFSCs

Interactions between mesenchymal cells and epithelial stem cells are crucial for morphogenesis of hair follicles. Hair follicle stem cells (HFSCs) have multidirectional differentiation potential, which can differentiate into skin, hair follicles, and sebaceous glands [45,46].The transcriptome analysis conducted by Chen et al. [47] demonstrated the various different expressed genes upon crosstalks between them that were enriched in a variety of pathways, among which is the PI3K/Akt pathway. The expression of various growth factors and cytokines, including FGFs, IL6, and oncostatin M that potentially activate PI3K/Akt pathway, were upregulated in both cell types. The results also showed that the pathway was significant to establish the interaction between the two cell types in the regeneration of hair follicle. Another study revealed that the PI3K/Akt pathway plays a vital role in the transformation of wounding-induced hair follicle telogen into anagen, and deficiency of Pten in Lgr5+ HFSCs cause the proliferation of stem cells, which contributes to hair follicle regeneration. The exosomes in platelets-rich plasma (PRP) were also demonstrated to promote hair follicle stem cells survival via the Akt/Bad cascade pathway [48]. The activation of the PI3K/Akt pathway is supposed to act as a promising therapeutic target related to hair regeneration, based on the available direct evidence of the function of PI3K/Akt in hair follicle regeneration and the potential role of Akt activation in PRP therapy. Long non-coding RNAs (lncRNAs), identified as non-coding transcripts (>200 nucleotides), are known to be essential for the differentiation and proliferation of various stem cells, including HFSCs [49]. CAI et al. [50] discovered that lncRNA5322 could target the miR-21-mediated PI3K-Akt signaling pathway to promote the proliferation and differentiation of HFSCs.

### 4.3. Activation of PI3K/Akt/mTOR Pathway Could Promote EMT Process and then Enhance Skin Wound Healing

Skin wound healing is a comprehensive and complex process involving inflammation response, new tissue formation, and tissue remodeling that consists of proliferation and migration of various cell types (inflammatory cells, keratinocytes, fibroblasts, platelets) to restore the integrity of the skin barrier [51,52,53,54,55,56]. It is already recognized that the PI3K/Akt pathway is strongly associated with the formation of an epidermis barrier that mainly depends on keratinocyte’s proliferation and differentiation. The study conducted by Chen et al. [57] demonstrated that miR-126 binding to its target gene PLK2 promotes the proliferation and migration of keratinocytes, thereby playing a significant role in skin wound healing via activation of the PI3K/Akt pathway. Jiang et al. [58] also indicated that miR-26a could reduce the migration of keratinocytes by regulating its target gene, ITGA5, thereby inhibiting wound healing. Epithelial–mesenchymal transition (EMT) is demonstrated to involve in skin wound healing and activation of PI3K is considered to activate the mTOR via Akt to accelerate the EMT [59,60,61]. Xiao et al. [62] also found that the treatment of ozone oil accelerated activation of the PI3K/Akt/mTOR pathway to promote the EMT process and then enhance wound healing. In future, the targeting therapies of the PI3K/Akt/mTOR pathway may offer new hope for skin wound healing.

### 4.4. PTEN/PI3K/Akt Pathway Functions in the Skin Senescence and Self-Renewal of hSKPs

The PI3K/Akt signaling pathway cascade is closely related to aging and lifespan regulation for the whole organism, since it can profoundly alter the activity and number of different types of stem cells [63,64]. The previous in vivo and in vitro investigations have demonstrated a pivotal role of the PI3K-Akt pathway in the neural stem/progenitor cells’ self-renewal and differentiation [65,66,67]. The skin-derived precursors (SKPs) were identified to have great value in the reconstitution of skin and hair follicle [68,69,70,71]. Liu et al. [72] discovered the significant role of the PI3K/Akt pathway in the senescence and self-renewal of hSKPs. Reactive oxygen species (ROS) is mainly derived from oxidative cell metabolism and is necessary for both chronological aging and skin photoaging [73,74]. The PTEN, a type of tumor suppressor, could dephosphorylate the lipid second messenger, phosphoribosyl 3,4,5-trisphosphate (PIP3), an enzymatic product of PI3K, acting as a negative element to regulate the survival signaling mediated by the PI3K/Akt pathway [75,76]. Noh et al. [77] discovered that downregulation of PTEN expression and following activation of PI3K signaling led to activation of PKC, which then increased ROS production via NADPH oxidase expression and its activity regulation in reproductive senescent HKFs.

### 4.5. Activation of PI3K/Akt Pathway Protects Melanocytes from Oxidative Stress

Melanocyte homeostasis is also an important part of skin homeostasis. Hyperproliferation of melanocytes lead to nevi melanoma and destruction of melanocytes is associated with depigmented skin disorders like vitiligo. Accumulative ROS induces melanocytes apoptosis and therefore promotes its destruction. Activation of the PI3K/Akt pathway protects the melanocytes from apoptosis induced by oxidative stress [78,79,80]. The Bcl-2 and caspase family both are crucial downstream molecules of the PI3K/Akt signaling pathway, participating in the apoptotic process induced by ROS. Increased expression of the anti-apoptotic protein, Bcl-2, and decreased expression of apoptotic proteins like Bax (also a member of Bcl-2 family) and caspases 3 and 9 are the result of activation of the PI3K/Akt pathway [81]. The effect is supposed to be reversed by the PI3K inhibitor LY294002 [82]. Additionally, the upstream element, overexpression of PTEN, could inhibit activation of the PI3K/Akt pathway and then lead to the melanocyte destruction or death [83]. However, excessive and continuous activation of the PI3K/Akt pathway induced by the down-regulated expression of PTEN might cause the occurrence and development of melanoma [84,85]. Taken together, normally activated PI3K/Akt pathway is crucial to maintain the melanocyte homeostasis.

## 5. Emerging Roles in Non-Malignant Skin Disorders

### 5.1. The PI3K/Akt Pathway Is Involved in the Formation of Acne by Inhibiting Lipogenesis

Acne vulgaris is a common recurrent inflammatory skin disease with a complicated mechanism [86,87]. The major hormones like androgens, insulin, and insulin-like growth factor-1(IGF-1) are responsible for the occurrence and development of acne vulgaris [88]. Insulin and IGF-1 activate the PI3K/Akt cascade, which upregulates the nuclear export of forehead box protein O1 (FoxO1), as a key component in the process of acne formation, antagonizing the expression of SREBP-1c and suppressing the transactivation of AR to inhibit lipogenesis [89,90,91]. Additionally, FoxO1 also elicits the activation of the adenosine 5′-monophosphate-activated protein kinase (AMPK) pathway to negatively regulate the mTORC1, another key downstream element of the PI3K/Akt pathway [92,93]. Taken together, insulin and IGF-1 both enhance lipid synthesis by regulating the attenuation of FoxO1 inhibition. Some existing drugs impact lipid synthesis via Akt/FoxO1/mTOR or AMPK signaling. For instance, epigallocatechin-3-gallate (EGCG) inhibits IGF-induced lipogenesis in SZ95 sebaceous grand cells by downregulating the level of mTOR and S6 ribosomal protein, which are both crucial downstream elements in the PI3K/Akt pathway [94]. EGCG also decreases the production of sebum by activating the AMPK-SREBP-1 pathway [93]. It is recognized that the oral isotretinoin is an excellent agent for most patients of severe recalcitrant acne; it has been approved by the FDA [95]. It promotes the apoptosis of sebaceous gland cells to reduce the production of serum, because it upregulates the nuclearFoxO1 and FoxO3 proteins expression [96]. It is known that tumor necrosis factor-alpha (TNF-a) is involved in the formation of acne and increasing cases has reported the efficacy of anti-TNF-a agents in the management of moderate to severe acne [97,98,99]. However, the definite mechanism is unclear. Choi et al. [100] clarified that TNF-a could induce the activation of SREBP-1 and increase lipogenesis via the PI3K/Akt pathway and c-jun N-terminal kinase (JNK) in human sebocytes. Moreover, they suggested that anti-TNF-a agents is supposed to become potential therapeutic strategies to control seborrhea, which is also a common skin inflammatory disease, characterized by excessive secretion of sebum by sebocytes, similar to acne.

### 5.2. Dysregulation of PTEN, FOXO, and mTOR of the PI3K/Akt Signaling Cascade Are Associated with the Occurrence of Psoriasis

Psoriasis is a commonly chronic inflammatory cutaneous disorder; one of its typical distinctions is excessive proliferation and abnormal apoptosis of keratinocytes [101]. The elements of the PI3K/Akt pathway cascade, including PTEN, FOXO, and mTOR, are all believed to involve the growth, survival, and proliferation of keratinocytes.

The mRNA expression and protein levels of PTEN in skin lesions of psoriasis were found to be decreased, compared with normal skin [102]. The overactivation of Akt induced by the decreased expression of PTEN could contribute to the psoriatic lesion by promoting the abnormal proliferation and apoptosis of keratinocytes [103]. While in normal skin, the normal PI3K/Akt pathway activation is crucial for cell proliferation and multiplication in the basal epidermis and for terminal differentiation in the upper layers [104].

Previous studies have indicated that the PI3K/Akt/mTOR pathway is highly expressed in human and murine psoriatic lesions. It is observed in psoriasis that PI3K binds to Akt and thereby in turn activites the mTOR to promote keratinocyte hyperproliferation and inhibite differentiation [105,106,107,108]. It has also been reported that the dysregulation of cytokines and growth factors like IL-17 and IL-12 could activated the mTOR pathway, which is an essential factor to regulate the proliferative and inflammatory process in psoriasis [109,110,111]. Additionally, the PI3K/Akt/mTOR pathway is a key upstream autophagy signal transduction pathway. Defects in autophagy induces the occurrence and development of psoriasis by promoting the production of inflammatory cytokines [112]. Recently, the PI3K/Akt/mTOR pathway was reported to function in Th1/Th2/Th17 imbalance, which is key in the occurrence and development of psoriasis [113].

It is recognized that overexpression of PI3K could induce excessive activation of AKT and then in turn phosphorylate the downstream target proteins like FOXO to promote cell proliferation. In the psoriatic keratinocytes, FOXO expression is mostly found in the cytoplasm but not in the nucleus in keratinocytes of uninvolved skin of psoriasis and normal skin [114]. Moreover, P-Akt, which induces the suppression of FOXO1 expression, is highly expressed in the psoriatic lesion, compared with normal skin [115]. Therefore, excessive activation of P-Akt might alter the cellar location of FOXO from the nucleus to the cytoplasm, losing the function to inhibit cell proliferation with keratinocyte hyperproliferation [116].

Since the PI3K/Akt cascade pathway is recognized in the pathogenesis of psoriasis, it is expected to be a promising anti-psoriatic target. Recently, topical rapamycin and delphinidin, a dietary antioxidant found abundantly in pigmented fruits and vegetables as therapeutic agents of psoriasis have been identified to alleviate psoriatic lesions in the imiquimod (IMQ)-induced psoriasis phenotype in mice via suppressing the PI3K/Akt/mTOR pathway [117,118,119]. The results of the study conducted by Yue et al. [120] demonstrate that PSORI-CM02 inhibited the proliferation of HaCaT cells by suppressing the autophagy process induced by the PI3K/Akt/mTOR pathway. Additionally, there are studies showing that matrine may inhibit keratinocytes proliferation via PI3K/Akt/FOXO signaling pathways and silibinin exerts its influences through negative regulation of PI3K/Akt pathways [121,122]. Taken together, it is expectable that targets of PI3K/Akt could be developed into successful agents to manage psoriasis.

### 5.3. The PI3K/Akt Pathway Is Involved in the Onset and Exacerbation of Atopic Dermatitis

Similar to psoriasis, atopic dermatitis (AD) is one of the most common chronic recurrent inflammatory cutaneous disorders related to T cell [123]. Xiao et al. [124] designed a study to explore the PI3K/Akt pathway activity in the peripheral T cells of patients with AD and its clinical value. They found that the PI3K/Akt pathway is aberrantly activated in peripheral T cells from patients of AD, and its activation corresponds to T cell proliferation and cytokine secretion like IL-6 and IL-10. Chronic AD is always presented with lichenified thickening lesion as the result of thickening epidermis and dermis. Moriya et al. [125] indicated that the IL-13, reported to be increased in the skin tissue of human AD, could downregulate the expression of MMP-13 both in the level of mRNA and protein to induce the thickened dermis by decreasing collagen degradation. However, IL-3 upregulate the MMP-13 expression via inhibiting the PI3K/Akt pathway.

### 5.4. The PI3K/Akt Signaling Pathway Is Associated with the Pathologic Fibrosis of Human Scleroderma

Scleroderma is mainly characterized by progressive dermal fibrosis, as a result of overexpression of profibrotic cytokines and excessive deposition of collagen [126]. Under the hypoxic condition, the PI3K/Akt/mTOR pathway cascade is activated through the Akt phosphorylation occurred at Ser473 and Thr308 sites and mTOR at Ser2448 site in scleroderma fibroblasts [127]. Zhou et al. [128] found that inhibitor of PI3K/Akt, LY294002, and the classic mTOR inhibitor rapamycin could both significantly suppress the HIF-1a up-regulation [129], CTGF [130], and collagen I, are all conceived as crucial cytokines involved in the dermal fibrosis of scleroderma, demonstrating that PI3K/Akt/mTOR/HIF-1a-mediated fibrogenesis was critically participated in the activation of SSc fibroblasts. The 2-methoxyestradiol (2-ME), a natural endogenous metabolite of 17b-estradio, reduced HIF-1a, CTGF, and collagen I induced by hypoxia via PI3K/Akt/mTOR/HIF-1a and inhibited the proliferation of fibroblasts, which is expectable to become a promising agent to treat scleroderma [131].

Periostin, a type of matricellular protein, is known to have various functions that may be associated with skin fibrosis [132]. Yang et al. [133] found that periostin could accelerate pathologic fibrosis in human scleroderma. Additionally, the results of their study demonstrated that rmPeriostin can in vitro accelerate mouse dermal fibroblasts proliferation, partially through the periostin-PI3K/Akt signaling pathway.

### 5.5. The PI3K/AKT Pathway Is Involved in the Formation of Keloid via Promoting the Proliferation, Migration, Expression Levels of EM-Related Proteins in HKFs

Distinct with hypertrophic scars (HS), keloids (KD) are characterized by high rates of recurrence after surgery and invasion, which coincides with the features of tumors. Human keloid fibroblasts (HKFs) are the major effector cells participating in the pathogenesis of keloids [134]. LV et al. [135] found that compared with human dermal fibroblasts (HDFs) and normal skin tissues, the expression levels of Runx2 were significantly upregulated both in HKFs and keloid tissues. The result of enrichment analysis of the KEGG signaling pathway demonstrated that the PI3K/Akt pathway is a major one in keloids, where differently expressed genes were mainly involved in. The phosphorylation levels of PI3K and Akt were significantly reduced after si-Runx2 transfection, suggesting that the PI3K/Akt pathway participated in maintaining the HFK functions. Runx2 deletion in HKFs inhibited cell proliferation, migration, expression levels of EM-related proteins via the suppression of the PI3K/Akt pathway activation. Zhu et al. [136] revealed that the expression trend of 26 miRNAs was transformed in KDs but not in HSs, where they discovered that miR-188-5p is probably involved in the enlargement and invasion of KDs. The expression and activity of MMP-2 and MMP-9 were downregulated while miR-188-5p was upregulated in KFs. They are both influenced by the PI3K/Akt pathway. The results of their study also demonstrated that after treating KFs with miR-188-5pmimic and transfecting HSFs with miR-188-5p inhibitor, the expression levels of PI3K and phosphorylated Akt proteins were significantly decreased in KFs, while the expression was enhanced after downregulating miR-188-5p in HSFs. Taken together, the miR-188-5p may influence the proliferation, migration, and invasion of scar fibroblasts through the PI3K/Akt/MMP-2/9 signal transduction pathway.

### 5.6. Activation of PI3K/Akt Protect the Melanocytes from Destruction in Vitiligo

Vitiligo is a multifactorial disorder characterized with depigmented lesions induced by melanocyte destruction [137]. Oxidative stress is a crucial element involved in the occurrence and development of vitiligo, because accumulative ROS causes an autoimmune response that results in the destruction of melanocytes [138], which could be inhibited by the activation of the PI3K/Akt pathway [80]. Zhu et al. [83] discovered vitiligo lesions with highly expression of PTEN and in turn decreased the Akt phosphorylation, that might elicit human melanocytes death. Mesenchymal stem cells (MSCs) could decrease the expression of PTEN and then activate the PI3K/Akt pathway to promote the proliferation of melanocytes and inhibit melanocytes damage from the ROS. Additionally, as a crucial downstream factor of the PI3K/AKT pathway, Nuclear factor erythroid 2-related factor2 (Nrf2) is a master transcription factor in cellular defense against oxidative stress, which is the key ROS-induced overexpression of antioxidant proteins. ROS are demonstrated to activate Nrf2 through the PI3K/Akt pathway, protecting cells from oxidative stress [139,140]. Kim et al. [141] indicated that impairment of PI3K activation of keratinocytes in vitiligo lesions are susceptible to apoptosis induced by ROS-generating chemicals owing to reduced Nrf2 activation. Wan et al. [142] also discovered that α-MSH-induced activation of mTORC1, a downstream element of the PI3K/Akt pathway, helps maintain the dendrites of melanocytes under the condition of oxidative stress.

### 5.7. Activation of the PI3K/Akt Pathway Is Related to the Inhibition of HFSCs Apoptosis in the Pathogenesis of Androgenic Alopecia

Androgenic alopecia (AGA) is a common type of hair loss, which is known as the result of interaction between 5a-DHT level and genetic predisposition [143]. Several researches have indicated the roles of HFTs in the onset of AGA [144,145]. However, not all types of HFTs but the specific CD200-rich and CD34-positive HFSCs are mainly in deficiency in the bald scalp of AGA patients [146]. As a key pathway associated with apoptosis of all types of cells, Zhang et al. [147] found that vascular endothelial growth factor (VEGF) could elicit Akt phosphorylation. The application of the PI3K inhibitor, LY294002, prevents CD200-rich and CD34-positive HFSCs from apoptosis induced by 5a-DHT. Therefore, it is supposed that VEGF prevents the target HFSCs from apoptosis via the PI3K/Akt pathway (Table 1).

## 6. Conclusions

The PI3K/Aktpathway is a crucial signaling transduction pathway for the processes of cell growth, proliferation, survival, cell apoptosis inhibition, lipogenesis inhibition, et al. For the skin, the signaling pathway is crucial for the survival, growth, proliferation, regeneration and apoptosis of keratinocytes, dermal fibrocytes, hair follicle stem cells etc. and maintenance of their functions. Therefore, PI3K/Akt is closely associated with the epidermal barrier function, hair follicle regeneration, skin wound healing and skin senescence. However, a dysregulated PI3K/Akt signaling pathway could induce a series of malignant or non-malignant skin disorders. Hence, it is important that the PI3K/Akt signaling pathway intensity is strictly mediated during skin generation, homeostasis, and development. For precise regulation of the PI3K/Akt pathway intensity, targeting regulators of PI3K/Akt signaling are necessary, along with interaction between the PI3K/Akt and other signal transduction pathways like AMPK pathway and mTOR. Therefore, it is supposed that fine-tuning the intensity of a PI3K/Akt signal pathway may participate in skin homeostasis. As mentioned above, aberrant activation of PI3K/Akt induces the occurrence of groups of non-malignant skin disorders including acne, psoriasis, atopic dermatitis, scleroderma, keloid, vitiligo and AGA. Thus, enhancing understanding the precise mechanisms of PI3K/Akt pathway regulation is essential to advance new therapeutic strategies to maintain skin homeostasis. Moreover, corresponding knowledge may also help to present new therapeutic strategies for non-malignant skin disorders mentioned above.

## Figures and Tables

**Figure 1 cells-10-01219-f001:**
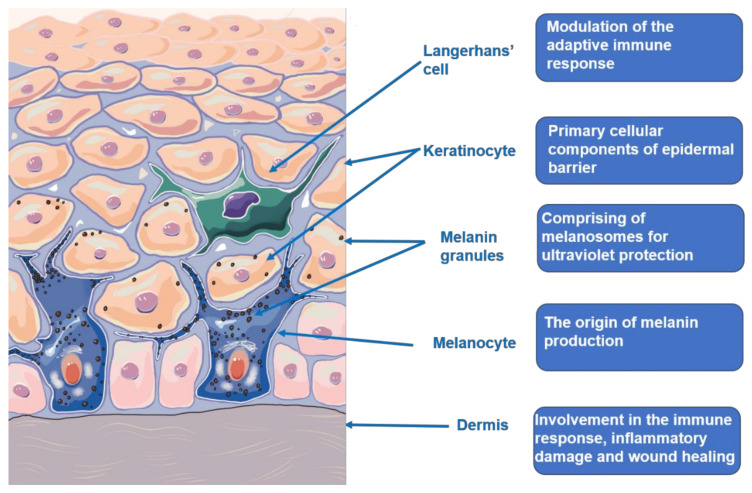
Skin microstructure and function.

**Figure 2 cells-10-01219-f002:**
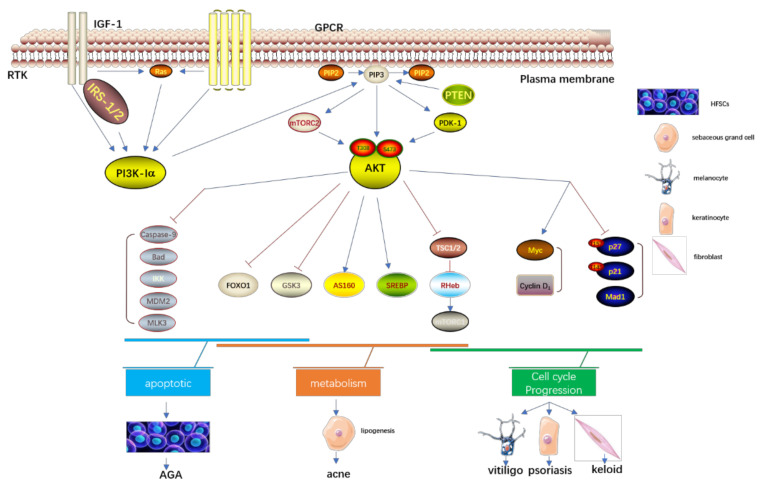
The PI3K/Akt signaling pathway and a group of non-malignant skin disorders; IGF-1, insulin-like growth factors-1; IRS-1/2,insulin receptor substrate-1/2; AKT, protein kinase B; IGF-1, insulin-like growth factors-1; IRS-1/2,insulin receptor substrate-1/2; FoxO1, Forkhead box O1; GSK3, glycogen synthase kinase 3; GPCR, G-protein-coupled receptors; IRS-1/2,insulin receptor substrate-1/2; IKK, IkB-kinase; MAD1, MAX dimerization protein 1; MDM2, murine double minute 2; MLK3, mixed lineage kinase 3; mTORC1/2, mTOR complex 1/2; PDK1, phosphoinositide-dependent protein kinase 1; PI3K, phosphatidylinositol 3-kinase; PIP2, phosphatidylinositol 4,5-biphosphate; PIP3, phosphatidylinositol 3,4,5-triphosphate; PTEN, Phosphatase and tensin homologue; RTK, receptor tyrosine kinases; SREBP, sterol regulatory element-binding proteins; TSC1/2, tuberous sclerosis complex 1/2; AGA, Androgenic alopecia.

**Table 1 cells-10-01219-t001:** PI3K/Akt pathway-related proteins and cytokines involved in non-malignant skin disorders.

Non-Malignant Skin Disorder	PI3K/Akt Pathway-Related Cytokines or Proteins
Acne	IGF-1, FoxO1, mTORC1, SREBP-1c, AMPK, TNF-a
Psoriasis	PTEN, IL-12, IL-17, mTOR, FOXO
Atopic dermatitis	IL-6, IL-10, IL-13, MMP-13
Scleroderma	Periostin, HIF-1a, CTGF and collagen I
Keloid	MMP-2, MMP-9
Vitiligo	PTEN, Nrf2, mTORC1
Androgenic alopecia	VEGF

## Data Availability

No datasets were generated or analyzed during the current study.

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
