# Peer review of "The PI3K/Akt Pathway: Emerging Roles in Skin Homeostasis and a Group of Non-Malignant Skin Disorders"

_cells, 2021, doi:10.3390/cells10051219_

Round 1
Reviewer 1 Report
This paper is a study on the mechanism of the PI3K/AKT Pathway in various skin diseases, and reviews a vast amount of content, and it is worth being published in this journal.
However, it would be better to add a picture of the role of PI3K/AKT in each skin disease. Please add additional pictures to show the relationship with other mechanisms.
Author Response
Dear reviewer 1:
Special thanks to you for your good comments. Considering your suggestion, we have modified the Figure 2 by adding the relationship between the diseases and PI3K/Akt pathway.
I sincerely hope this correction will have your recognition.
Yan Teng
Yours sincerely
Reviewer 2 Report
1) The link of the JNK and PI3K/Akt pathways should added and discussed in more detail.
2) Paragraph 3, more information on forkhead box O (FOXO) protein should be added for underlying its important function role.
3) Figure 2 should be split in two parts (A and B) with two separate legends and details. Also, please add abbreviations in figure’s legend.
4) The abbreviation of the PI3K/Akt appears in two different formates (as PI3K/Akt and as PI3K/AKT). The authors should be consisted with abbreviations.
5) A short discussion of the connection of the PI3K/Akt with the Nrf2 Signaling Pathway can be also added.
Author Response
Dear reviewer2:
Thank you for your comments concerning our manuscript. These comments are all valuable and very helpful for revising and improving our review. We have studied comments carefully and have made corrections which we hope meet with approval. Revised portion are marked in red in the paper. The main corrections and the responds to the reviewer's comments are as following:
Question 1: The link of the JNK and PI3K/Akt pathways should added and discussed in more detail.
Response: The link of the JNK and PI3K/Akt pathway has been inserted into the 5.1 paragraph as the last second sentence.
Question 1: Paragraph 3, more information on forkhead box O (FOXO) protein should be added for underlying its important function role.
Response: We have added more information on FOXO protein in the paragraph 3 that are marked in red.
Question 3: Figure 2 should be split in two parts (A and B) with two separate legends and details. Also, please add abbreviations in figure’s legend.
Response:I have modified the Figure 2 and add the abbreviations in figure’s legend.
Question 4: The abbreviation of the PI3K/Akt appears in two different formates (as PI3K/Akt and as PI3K/AKT). The authors should be consisted with abbreviations.
Response: I have unified the format as the "PI3K/Akt". The two format are both recognized in the previous literatures.
Question 5: A short discussion of the connection of the PI3K/Akt with the Nrf2 Signaling Pathway can be also added.
Response: we have added the connection of the PI3K/Akt with the Nrf2 into the 5.6 paragraph.
We appeciate for your warm work earnestly, and hope these corrections will meet with approval.
Reviewer 3 Report
The subject is of interest. However, an in depth analysis with appropriate references is expected from review paper. The citations are in many cases random and not representative. Please follow the recommendations outlined by Dr Blaustein, previous editor of Endocrinology (Endocrinology, 151(1):1–3, 2010)
Specific comments
In the abstract the authors focus on melanoma and bcc, while the review covers different malignant and non-malignant disorders. Please reconcile. Also why SCC is not mentioned.
Focus in the review on selected disorders in more detailed fashion is suggested with focus on diseases where PI3K/AKT Pathway's role is well appreciated. I suggest non-malignant and malignant disorders.
Fig. 1 is a textbook picture not necessary for expert review, unless function is added.
When mentioning sensory and regulatory function of the skin mention its role in regulation of local homeostasis (Sensing the environment: Regulation of local and global homeostasis by the skin neuroendocrine system. Adv Anat, Embryol Cell Biol 212: 1-115, 2012) and their neuroendocrine capabilities (Endocrine Rev 21, 457-487, 2000; Physiol Rev 80, 979-1020, 2000; Endocrine Rev 34:827-884, 2013).
Mention diverse reactions to ultraviolet light (Endocrinology 159(5), 1992-2007, 2018).
When discussing barrier function cite papers by Elias et al from UCSF.
Sections on melanocytes and pigmentation are suboptimal and should include hormonal and nutritional regulations of their functions (Physiol Rev 84, 1155-1228, 2004; Pigment Cell Melanoma Res 25, 14-27, 2012).
Role of melanogenesis and melanin pigment in regulation of metabolism should be mentioned (Arch Biochem Biophys 563:79-93, 2014; Exp Dermatol 24: 258-259, 2015)
Discussion of vitiligo is suboptimal.
Better figures illustrating the concepts presented are expected
Author Response
Dear reviewer 3:
I quitely appreciate your insightful comments. I found these comment are very helpful and I learned a lot by studying the literature you recommended. The revisions are addressed point by point below. Hope you could be satisfied with them.
Comment 1: In the abstract the authors focus on melanoma and BCC while the review covers different malignant and non-malignant disorders. Please reconcile. Also why SCC is not mentioned.
Response: In the abstract, I just want to simply display the malignant skin disorders associated with the PI3K/Akt pathway. According to your suggestion, I add the "SCC" that may be more comprehensive.
Comment 2: Focus in the review on selected disorders in more detailed fashion is suggested with focus on diseases where PI3K/AKT Pathway's role is well appreciated. I suggest non-malignant and malignant disorders.
Response 2: The roles of PI3K/Akt pathway has reviewed in the Cells, 2019, 8(8):803-. Therefore, I mainly reviewed the roles in the non-malignant skin disorders. Additionally, I have modified one of the words in the title from the "benign" to "non-malignant" that may be consistent with the content of paper.
Comment 3: Fig. 1 is a textbook picture not necessary for expert review, unless function is added.
Response: I have modified the figure1 and described the function.
Comment 4: When mentioning sensory and regulatory function of the skin mention its role in regulation of local homeostasis (Sensing the environment: Regulation of local and global homeostasis by the skin neuroendocrine system. Adv Anat, Embryol Cell Biol 212: 1-115, 2012) and their neuroendocrine capabilities (Endocrine Rev 21, 457-487, 2000; Physiol Rev 80, 979-1020, 2000; Endocrine Rev 34:827-884, 2013).
Response:According to your suggestions, I have added the content into the paragraph "The skin stucture and function" and cited the references(14-17).
Comment 5: Mention diverse reactions to ultraviolet light (Endocrinology 159(5), 1992-2007, 2018).
Response: According to your suggestions, I have added the content into the paragraph "The skin stucture and function" and cited the reference(18).
Comment 6: When discussing barrier function cite papers by Elias et al from UCSF.
Response: In the paragraph"The skin stucture and function", I have cited the papers in the reference(19.20)
Comment 7: Sections on melanocytes and pigmentation are suboptimal and should include hormonal and nutritional regulations of their functions (Physiol Rev 84, 1155-1228, 2004; Pigment Cell Melanoma Res 25, 14-27, 2012).
Response: According to your suggestions, I have added the related content into the paragraph "The skin stucture and function" and cited the reference(23-24).
Comment 8: Role of melanogenesis and melanin pigment in regulation of metabolism should be mentioned (Arch Biochem Biophys 563:79-93, 2014; Exp Dermatol 24: 258-259, 2015)
Response: I have added the related content into the paragraph"The skin struction and function" and cited the reference(25-26), which is marked in red.
Comment 9: Discussion of vitiligo is suboptimal
Response: I have rewritten this part in the paragraph" The skin struction and function" which is marked in red.
Comment 10: Better figures illustrating the concepts presented are expected
Response: I have modified the Figure1 and Figure 2 according to your suggestion.
I again appreciate for your warm and careful work earnestly, and hope the corrections will meet with approval.
Round 2
Reviewer 3 Report
The authors made significant corrections which is appreciated.
There are some minor corrections to be performed
There is a problem with reference formatting. For example following references should corrected.
Reference 14 should read as:
Slominski A, Zmijewski MA, Skobowiat C, Zbytek B, Slominski RM, and Steketee JD (2012). Sensing the environment: Regulation of local and global homeostasis by the skin neuroendocrine system. Adv Anat, Embryol Cell Biol 212: 1-115.
Reference 15 should read as
Slominski A, Wortsman J (2000) Neuroendocrinology of the skin. Endocrine Rev 21, 457-487.
Reference 16 should be replaced by
Slominski AT, Manna PR, Tuckey RC (2015) On the role of skin in the regulation of local and systemic steroidogenic activities. Steroids 103, 72-78.
Reference 23 should read as:
Slominski A, Zmijewski, M, Pawelek J (2012) L-tyrosine and L-dihydroxyphenylalanine as hormone-like regulators of melanocyte functions. Pigment Cell Melanoma Res 25, 14-27
Reference 25 should read as:
Slominski A, Kim TK, Brozyna AA, Janjetovic Z, Brooks DL, Schwab LP, Skobowiat C, Jozwicki W, Seagroves TN (2014) The role of melanogenesis in regulation of melanoma behavior: Melanogenesis leads to stimulation of HIF-1α expression and HIF-dependent attendant pathways. Arch Biochem Biophys 563:79-93.